# Deforestation Detection in the Amazon Using DeepLabv3+ Semantic Segmentation Model Variants

**Renan Bides de Andrade** [†]**, Guilherme Lucio Abelha Mota** *,[†] **and Gilson Alexandre Ostwald Pedro da Costa** [†]

Post-Graduation Program in Computational Sciences, Rio de Janeiro State University (UERJ),
Rio de Janeiro 20550-013, RJ, Brazil
* Correspondence: guimota@ime.uerj.br
† These authors contributed equally to this work.

**Abstract:** The Amazon rainforest spreads across nine countries and covers nearly one-third of South America, being 69% inside Brazilian borders. It represents more than half of the remaining tropical forest on Earth and covers the catchment basin of the Amazon river on which 20% of the surface fresh water on the planet flows. Such an ecosystem produces large quantities of water vapor, helping regulate rainfall regimes in most of South America, with strong economic implications: for instance, by irrigating crops and pastures, and supplying water for the main hydroelectric plants in the continent. Being the natural habitat of one-tenth of the currently known species, the Amazon also has enormous biotechnological potential. Among the major menaces to the Amazon is the extension of agricultural and cattle farming, forest fires, illegal mining and logging, all directly associated with deforestation. Preserving the Amazon is obviously essential, and it is well-known that remote sensing provides effective tools for environmental monitoring. This work presents a deforestation detection approach based on the DeepLabv3+, a fully convolutional deep learning model devised for semantic segmentation. The proposed method extends the original DeepLabv3+ model, aiming at properly dealing with a strong class imbalanced problem and improving the delineation quality of deforestation polygons. Experiments were devised to evaluate the proposed method in terms of the sensitivity to the weighted focal loss hyperparameters—through an extensive grid search—and the amount of training data, and compared its performance to previous deep learning methods proposed for deforestation detection. Landsat OLI-8 images of a specific region in the Amazon were used in such evaluation. The results indicate that the variants of the proposed method outperformed previous works in terms of the F1-score and Precision metrics. Additionally, more substantial performance gains were observed in the context of smaller volumes of training data. When the evaluated methods were trained using four image tiles, the proposed method outperformed its counterparts by approximately +10% in terms of F1-score (from 63% to 73%); when the methods were trained with only one image tile, the performance difference in terms of F1-score achieved approximately +18% (from 49% to 67%).

**Keywords:** DeepLabv3+; semantic segmentation; Amazon; deforestation detection

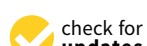



## 1. Introduction

The Amazon rainforest covers an area of approximately 5.5 million km², which amounts to approximately one-third the size of the South American continent. It spreads across nine countries, establishing an arc that begins in Bolivia and extends to French Guiana, being approximately 69% inside Brazilian borders. Altogether, the Amazon represents more than half of the remaining tropical forest area on Earth [1] and contains more than 10% of all the above-ground biomass on the planet [2]. The forest also holds about 20% of the total carbon sequestered by the world's terrestrial forests [3].

The forest covers the Amazon river catchment basin on which 20% of all free-flowing fresh water on Earth [4] flows. The dominant conjunction of the Amazon rainforest and river basin produces large quantities of water vapor reaching out vast extents in South

America. The so-called "flying rivers", generated through evapotranspiration by the Amazon ecosystem, carry out moisture to the north, west, and south. Thus, it helps to regulate rainfall regimes in most of South America [5]. The portion of moisture that heads south is also of enormous economic importance for irrigating crops and pastures, and for supplying water for the main hydroelectric plants on the continent.

Additionally, the biome contains unparalleled biodiversity, being the natural habitat of one-tenth of the currently known species [6], the largest concentration of plants and animal species in the world. Such biodiversity forms an extremely complex and large-scale ecosystem whose evolution naturally produced countless biological solutions. Currently, the vast majority of those species are still far from being sufficiently studied to ensure the exploitation of such enormous biotechnological potential.

Unfortunately, for many decades, the Amazon has faced several threats as a result of unsustainable economic development. Among the main menaces to the biome are the expansion of agricultural activities at an industrial scale, especially soybean cultivation and cattle raising, forest fires, and illegal mining and logging [7–9]. All those factors are directly associated with deforestation. Deforestation in the Brazilian Amazon also has important social and humanitarian consequences. While trying to fight for the protection of the forest for future generations, environmentalists [10,11] and members of traditional indigenous communities [12] are often targets of attacks by loggers [13], land grabbers or illegal fisherman, hunters, and miners.

The above-mentioned facts indicate the importance of the preservation of the Amazon forest, and it is well known that Remote Sensing (RS) data provide key capability for large-scale, low-cost, and risk-free environmental monitoring [14]. According to the Brazilian National Institute for Space Research (INPE) [15], deforestation accelerated significantly in the Brazilian Legal Amazon (BLA) area during the 1990s and early 2000s. Unfortunately, since 2019, it has reemerged with substantial intensity [16,17]. It is an urgent situation; the World Wildlife Fund [18] estimates that more than a quarter of the rainforest may vanish by 2030 if the current rate of deforestation continues. Important consequences of deforestation can already be observed; a recent report paper [19] asserts that between 2010 and 2019 Brazil's Amazon carbon stock had been in deficit, since it gave off 16.6bn tonnes of carbon dioxide, while was able to draw down only 13.9bn tonnes.

Since the late 1980s, INPE is in charge of monitoring deforestation in the Brazilian Legal Amazon. Among its main monitoring projects are PRODES [20], DETER-A [21] and DETER-B [22], which are mainly based on visual interpretation of RS data. The main reason for such a human-intensive approach is the accuracy requirement necessary for an official, governmental system. There is, therefore, a high demand for automatic methods that can reach the desired expectations in terms of accuracy values, lessening human intervention, while reducing response times.

Deforestation detection can be posed as a change detection problem. By and large, for identifying changed or unchanged pixels, there must be a pair of coregistered images for the same geographic location, acquired at different dates. Considering $t_i$ as the acquisition date of an image and $t_{i+d}$ as the date of an image from the same location acquired more recently, the problem to be tackled basically represents finding differences in pixel positions of both images associated with a level of forest suppression that can be regarded as deforestation.

Training a classifier for this task is a challenging problem. The first issue to be tackled is the intraclass variability of the unchanged class (i.e., *no-deforestation*), which encompasses forest pixels that remain standing at $t_{i+d}$, and of the changed class (i.e., *deforestation*). The deforestation pixel clusters may differ substantially depending on the deforestation practices in the region, which may be associated with clear-cutting, selective logging, or degradation through forest fire. Additionally, the texture in a dense forest may differ because of the sun position or other seasonal aspects [23]. The other important difficulty of the target application is the high unbalance between unchanged and changed class samples. Regardless of the deforestation rates, the occurrence of deforestation pixels is still at least one order of magnitude lower, in relation to the amount of no-deforestation pixels.

Despite the success of deep learning methods in many application fields, deep neural networks (DNNs) are known to require large amounts of labeled data to be properly trained [23]. That can be a serious problem for operational applications, especially those based on remote sensing, for which obtaining such reference data is usually costly and time-consuming. In the case of the deforestation detection application, for example, creating such reference data involves highly trained specialists, acquainted with the characteristics of the forest cover through time, and with the different spatial and spectral deforestation patterns. The selection of a proper DL model for RS applications should, therefore, take into consideration its demand for training data.

We hypothesize that the problems mentioned in the previous paragraphs, i.e., the intraclass variability, high class unbalance, and high demand for training data, can be successfully dealt with by selecting a proper deep learning architecture and a proper training procedure. Specifically, we believe that a fully convolutional, dense labeling architecture may lessen the intraclass variability issue, once it can learn not only specific class representations but also interrelationships among the classes of interest. Additionally, a proper design of a loss function, i.e., which gives higher importance to hard-to-classify samples, may help cope with intraclass variability and class unbalance and still lessen the demand for training data.

In this work, we propose a deep learning method based on a fully convolutional model, namely, the DeepLabv3+ [24]. In a previous publication [25], we presented preliminary results of this research. In the present work, we bring a thorough evaluation of the full set of experiments carried out afterward. In the course of this research, we adapted the DeepLabv3+ model to deforestation detection, by setting its input as a tensor formed by stacking the bands of two coregistered images. We also adjusted many of the model's hyperparameters, aiming at improving the delineation accuracy of deforestation polygons. Additionally, we substituted the original DeepLabv3+ cross-entropy loss function by the weighted focal loss function [26] and fine-tuned its parameters.

We also compared the results obtained with the proposed method with those reported in a previous work [27] over the same study area. Besides comparing the respective deforestation detection accuracies, we devised experiments to assess the methods' demands for training data. In short, the major contributions in this work are:

- Adaptation of the DeepLabv3+ semantic segmentation model to deforestation detection.
- Thorough grid search on the loss function hyperparameters for assessing their relative impact on deforestation mapping accuracy.
- Evaluation of the proposed method's sensitivity to the amount of training data.
- Extensive performance evaluation of the proposed method on an area of the Amazon forest.
- Comparison of the proposed method with previous deep-learning-based methods for deforestation detection.

The remainder of our article is organized as follows. We first review related works. In Section 3, we describe the evolution of the lineage which brought about DeepLabv3+. In Section 3, we present the adaptations of the original DeepLabv3+ model carried out in this research. Section 4 is dedicated to the description of the experimental procedure, and, in Section 6, we present and analyze the experimental results. We end the manuscript by bringing about the conclusions and briefly discussing directions for further work.

## 2. Related Works

According to Volpi and Tuia [28], deep learning (DL) approaches for image classification on a pixel level can be subdivided into two main approaches: patchwise classification and semantic segmentation. Broadly speaking, considering an image patch, patchwise classification produces a single decision for its central pixel. In contrast, semantic segmentation, given an image patch, produces at once concurrent decisions for each of its pixels.

By and large, remote sensing (RS) change detection (CD) approaches can be comprehended as an extension of pixel-level image classification. The main difference resides

in the fact that conventional image classification takes data from one time instance and provides a class label for each image pixel. On the other hand, CD relies on two or more coregistered images taken at distinct time instances, and classifies image pixels into changed or unchanged areas, sometimes attributing a semantic label to the changed pixels.

Dealing with change detection through deep learning leads to three main approaches. The first one is postclassification, in which the images from different epochs are classified, and afterward the classifications are compared to produce the change map. The second approach involves architectures containing two encoders, the so-called Siamese networks [29], where each encoder processes an image of the same region acquired at a different time instance. Finally, in the so-called early fusion approaches, image features acquired at different dates are combined before being forwarded to a single encoder network. We briefly describe below notable DL-based studies dedicated to deforestation detection.

Post-classification approaches based on DL are not so common. Zulfiqar et al. [30] exploited Landsat images for deriving a postclassification, semantic segmentation approach based on the U-Net architecture. The method was applied for forest estimation and change detection in Pakistan.

A number of early fusion patchwise classification approaches can be found in the literature. Khan et al. [31] used a variant of the VGG-16 architecture and a long time series of Landsat images for forest change detection in Australia. Masolele et al. [32] evaluated several deep learning approaches for land-use classification following deforestation using a large-scale dataset of Landsat images from Latin America, Africa, and Asia, where forests were converted to distinct uses. The approaches evaluated encompass 2D-CNN, LSTM, 3D-CNN, Hybrid CNN-LSTM, ConvLSTM, and CNN-MHSA. In a deforestation mapping effort in Ukraine, Shumilo et al. [33] compared LSTM and MLP for deforestation detection using Sentinel-1 and Sentinel-2 images.

Approaches based on semantic segmentation have recently become more prevalent in the literature. Lee et al. [34] compared semantic segmentation results provided by U-Net and SegNet for land-use/land-cover classification of high-resolution Kompsat-3 satellite images in Korean regions affected by deforestation. Isaienkov et al. [35] compared the performance of different semantic segmentation approaches for deforestation detection in the Ukrainian Forest Ecosystem. The authors exploited a dataset of Sentinel-2 multispectral images for the evaluation of both early fusion and dual branch networks, comprising seven approaches inspired by U-Net, Siamese networks, and LSTM.

Some recent remote sensing image classification methods, although not yet applied to deforestation detection, are worth mentioning due to their differentiated performance. Jia et al. [36] proposed a graph-in-graph (GiG) model and a related convolutional network (GiGCN) dedicated to HSI classification from a superpixel viewpoint. Some of such core ideas remount to Geobia approaches [37], while GiG representation covers information inside and outside superpixels, corresponding to the local and global characteristics of geo-objects. Meanwhile, the external graph is constructed according to the spatial adjacent relationships among superpixels. The approach extracts hierarchical features and integrates them into multiple scales, improving the discriminability of GiGCN. In addition, ensemble learning is incorporated to further boost method robustness. In order to classify hyperspectral images, Ahmad et al. [38] combine 3-D and 2-D inception networks with an attention mechanism. The resulting attention-fused hybrid network relies on three attention-fused parallel hybrid subnets with different kernels; and uses high-level features to enhance the final prediction maps. That approach is able to filter out the critical features for classification. Hong et al. [39] proposed a general multimodal deep learning framework aiming at overcoming the difficulties of DL in finely classifying complex scenes, due to the limitation of information diversity. The method is additionally able to explore the cross-modality inherent to a number of RS image classification applications. That framework, besides being able to perform pixel-wise classification, is also applicable to spatial information modeling with convolutional neural networks.

Several DL approaches specifically dedicated to deforestation detection in the Amazon were recently presented. De Bem et al. [40] evaluated SharpMask, U-Net, and ResU-net semantic segmentation algorithms for deforestation Detection. The proposed approach employs early fusion of Landsat images. Maretto et al. [41] adapted the U-Net architecture for deforestation detection using Landsat images. The authors compared early fusion and dual branch approaches, and created spatiotemporal variations of the U-Net architecture, which make it possible to incorporate both spatial and temporal contexts. Tovar et al. [42] evaluated the use of both spatial and channel attention models for deforestation detection. Watanabe et al. [43] use FCNs to build an early fusion semantic segmentation approach for deforestation detection in Brazil and Peru using PALSAR-2/ScanSAR images. Taquary et al. [44] propose a combination of LSTM and CNN for deforestation detection based on orbital SAR time series of Sentinel-1 images. The approach consists of an early fusion, patchwise classification technique. Shumilo et al. [45] exploits Sentinel-2 and Sentinel-1 imagery for deforestation detection. The approach employs U-Net-based variants, and a semisupervised learning technique. Experiments were carried out on a dataset of the Kyiv region. Torres et al. [46] evaluated several fully convolutional networks architectures (U-Net, ResU-Net, SegNet, FC-DenseNet, and DeepLabv3+) for deforestation detection. The performance of such state-of-the-art models was evaluated using two Brazilian Amazon datasets with different spatial and spectral resolutions (imagery from Landsat-8 and Sentinel-2). ResU-Net consistently outperformed the other methods, providing F1-scores of 70.7% for the Landsat dataset, and 70.2% for the Sentinel-2 dataset.

Adarme et al. [27,47] presented an evaluation of patchwise classification methods for automatic deforestation detection in the Amazon and Cerrado biomes. The compared approaches were both early fusion (SVM, CNN, CSVM) and dual branches networks (Siamese networks). The evaluated DL-based approaches outperformed the SVM baseline, both in terms of F1-score and overall accuracy. Among those works, [47] presented the most thorough experimental evaluation. The herein presented work follows the same experimental protocol in order to make a thorough comparison with those state-of-the-art approaches. Among the approaches presented in [47], the Early Fusion (EF) CNN and Siamese CNN (S-CNN) were used as experimental benchmarks for the evaluation that will be presented in the next sections.

The importance of [47] for deforestation detection is corroborated by its developments, as is the case of the present research. Soto et al. [23] represents another unfolding of [47]. That work presents a DL-based approach for domain adaptation in the context of change detection tasks. Such model adapts DANN [48] as well as the EF approach of [47]. That domain adaptation approach has shown to be able to improve the accuracy of cross-domain deforestation detection, considering different sites in the Amazon and Brazilian Cerrado biomes.

## 3. Evolution of the DeepLab Family Design

DeepLab is a semantic segmentation CNN family of models, which evolution aggregated several consistent breakthroughs. In this description, we follow its timeline to present the sequence of main ideas behind its evolution.

The elder DeepLab version was proposed in [49]; and brought among its major contributions a particular implementation of the "hole algorithm" originated to efficiently compute the undecimated wavelet transform [50]. In its first release, the so-called *atrous* convolution came about for preventing the network's output stride to go beyond the target value of eight. The DeepLabv1 encoder was built on top of the VGG-16 [51] architecture. Among the proposed adaptations were: (1) turning fully connected layers into convolutional ones (being the first an atrous convolution with a rate of four), (2) removing the last two pooling layers, and (3) converting the last three VGG-16 original convolutional layers into atrous convolutional ones with a dilation rate of two.

It should be stressed atrous convolutions—expressed by a regular, sparse linear transfer function—can considerably expand their receptive fields in relation to traditional convo-

lutional filters, without enlarging the number of parameters and computational complexity. They prevent the network's output stride from impacting the feature maps spatial resolution as a conventional sequence of convolutional followed by pooling blocks would do. The authors argue that the bigger the output stride the lesser accurate would be the silhouettes of the segmented objects.

The second version of DeepLab adopted the Spatial Pyramid Pooling (SPP) concept, which was introduced in the SPP-net architecture [52]. DeepLabv2 [53] proposed an atrous convolution-based SPP, the so-called ASPP, which comprises several parallel branches of atrous convolutions, with different receptive fields, coming together in a single set of feature maps. Therefore, such set of feature maps is capable of capturing information at various scales [53]. Thus, in comparison to the first release, DeepLabv2 is more efficient in segmenting objects at multiple scales. In addition, in its second release, the VGG-16's inspired DeepLab comes along with a residual net variant built on top of the ResNet architecture [54].

Until its second version, DeepLab relied on fully connected Conditional Random Fields (CRF) [55] to improve semantic segmentation outcomes. By and large, the CRF processes the outcomes of the deep convolutional neural network to provide the final output map. The use of CRFs was discontinued in the third DeepLab release [56]. As a surrogate for CRF, DeepLabv3 uses a naive decoder which performs bilinear interpolation of the output probability maps. As for the encoder, the authors experimented wit variants using backbones based on both VGG-16 and ResNet. The resulting features pass through an updated ASPP version which includes $1 \times 1$ filters, which stand for a degeneration of the high rate atrous filter that avoids image border effects, and image-level features, also known as image pooling [57]. Thus, the ASPP models multiple levels of context, gathering from local to global features.

Finally, the DeepLabv3+ model [24] adopted an encoder–decoder architecture with a DeepLabv3-like encoder structure. The proposed alternatives were based on ResNet-101 and Xception [58] backbones. Furthermore, a simple decoder module was devised aiming at enhancing segmentation results, especially alongside object boundaries. On one hand, the final features produced by the backbone pass through the ASPP to produce multiscale feature maps, which are then combined by a $1 \times 1$ convolution block. In such a context, the $1 \times 1$ convolution block reduces the number of channels associated with those features, so that they do not outweigh the importance of the encoder's output. On the other hand, low level backbone features go directly to the decoder to, then, be submitted to a $1 \times 1$ convolution block, whose result is concatenated with an upsampled version of the ASPP outcome after being submitted to its respective $1 \times 1$ convolution block. Such concatenated features are then submitted to a $3 \times 3$ convolution block that refines those features. Finally, to create the network output, such features are submitted to a simple bilinear upsampling module.

When adapting the Xception model [58] as the backbone of the encoder network for semantic segmentation, the authors used a deeper model like [59]. Max-pooling operations were replaced by depthwise separable convolution with striding, and batch normalization [60] and ReLU activation were added after each $3 \times 3$ depthwise convolution, as in the MobileNet design [61].

## 4. DeepLab-based Change Detection Method

In this section, we describe the adaptations performed on the DeepLabv3+ model to create the hereinafter called DeepLab-based Change Detection (DLCD) model. As the first amendment, it adopts the early fusion approach, taking as input a synthetic image created through stacking along the spectral dimension two coregistered images from different epochs. As a consequence of being based on an FCN, DLCD delivers dense labeling of the input patches.

Still considering the DLCD input layer, its design was also impacted by the current BLA deforestation circumstances. Indeed, in the dataset exploited for this work, deforestation

areas (or objects) are small and sparse, leading to large class unbalance, being the vast majority of the training set pixels either no-deforestation or unchanged. It should be stressed that such problem contours are dissimilar to the realty of PASCAL VOC 2012 and Cityscapes, the datasets used in [24]. So, considering such dissimilarities, we experimented with patches of smaller sizes than in [24]. After a preliminary evaluation, we obtained better and more consistent results using a patch size of $64 \times 64$ pixels. In terms of the DLCD backbone, we opted for the Xception architecture, since it provided the best results in [24].

As a consequence of such a small input patch, we opted for an encoder output stride of 8, the smallest evaluated in the original DeepLabV3+ [24]. In order to adjust the network architecture to the selected input patch size and output stride, we changed the dilation rates of the atrous convolutions in the ASPP to 3 and 6 (originally, dilation rates of 12 and 24 were employed), and removed the atrous convolution with the highest rate because it would degenerate into a $1 \times 1$ convolution. Considering the Xception backbone, to achieve an output stride of 8, we used only 3 convolutions with stride of 2 in entry flow, and only convolutions with stride of 1 in the middle and exit flow. Figure 1 shows the architecture of the proposed DLCD model.

As mentioned before, deforestation detection problems are usually characterized by a large class unbalance, where, by and large, changed pixels are orders of magnitude less numerous than unchanged ones. Thus, cross-entropy, $-\log(p_t)$, normally exploited in image classification problems, tend to fail in leading to consistent network weights for detecting changes. The main reason is that in dense labeling problems, it is generally not possible to force training set class balancing by simply replicating the less common class examples.

One alternative for dealing with such scenarios is the so-called weighted focal loss function (WFL) [26]. WFL is presented in Equation (1), where $y \in \{-1, 1\}$ specifies the ground-truth class label for which $y = 1$ indicates deforestation, while $y = -1$ means unchanged pixel; $p \in [0, 1]$ is the deforestation probability estimated by the proposed model's for a specific pixel; and $\alpha$ represents the weight associated to the deforestation class.

$$WFL(p_t) = -\alpha_t(1 - p_t)^\gamma \log(p_t)$$

$$\text{where} \quad p_t = \begin{cases} p & \text{if } y = 1 \\ 1 - p & \text{otherwise} \end{cases}$$

$$\text{and} \quad \alpha_t = \begin{cases} \alpha & \text{if } y = 1 \\ 1 - \alpha & \text{otherwise} \end{cases} \tag{1}$$

Equation (1) may be subdivided in three parts: (1) cross-entropy ($-\log(p_t)$); (2) weight relative to each class ($\alpha_t$); and (3) the term ($(1 - p_t)^\gamma$) that reduces the loss magnitude for well-classified pixels while enhances the loss of misclassified pixels, whereas a larger $\gamma$ intensifies such effect.

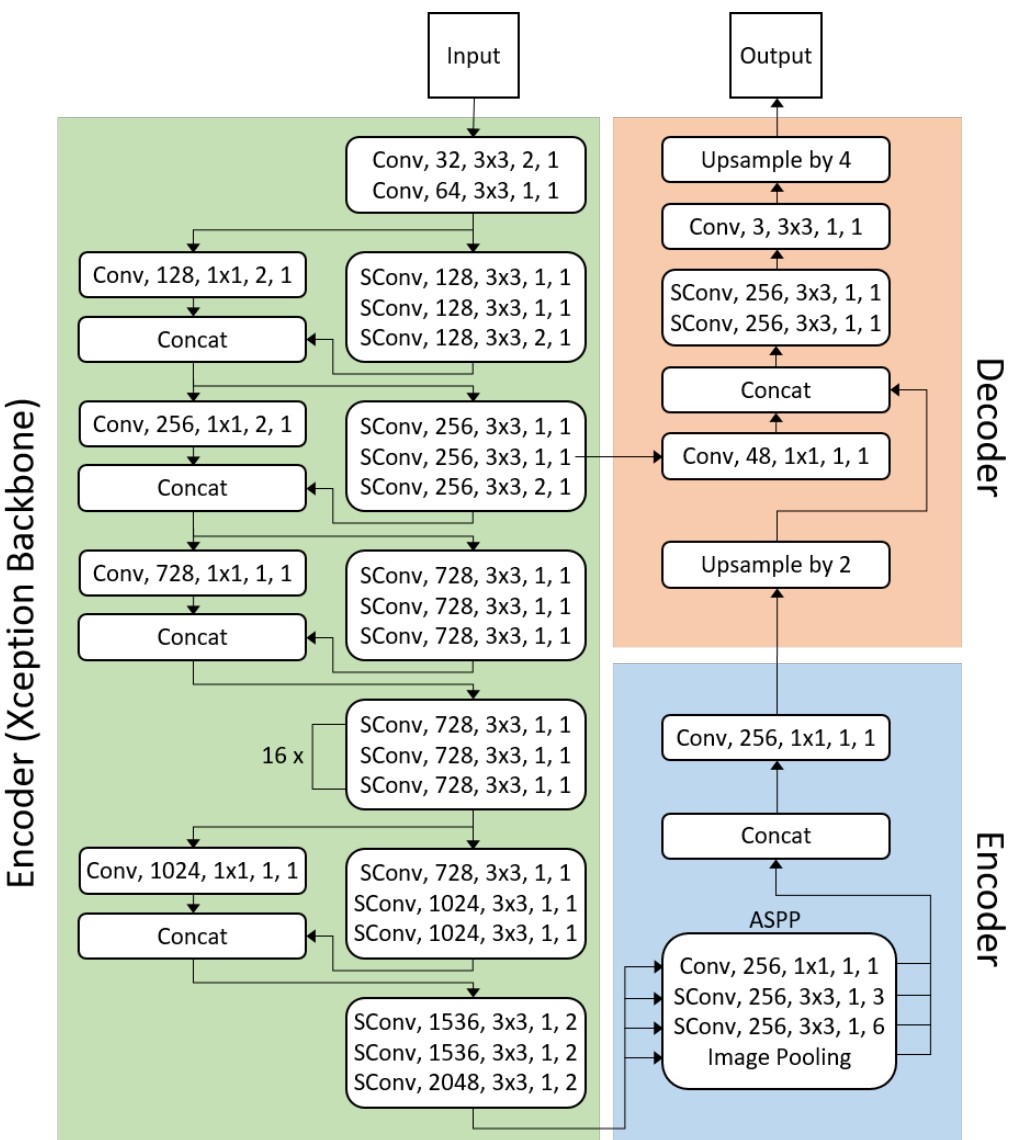

**Figure 1.** DLCD model diagram. The full model has a total of 40,639,856 parameters to be estimated. Layer descriptions contain: convolution type (Conv for regular convolution; SConv for depthwise separable convolution), number of filters, filter size, stride, dilation rate.

## 5. Experiments

In this section, we describe the dataset and the experimental setup including the approaches to be compared to DLCD.

### 5.1. Data Set Description

The reference area is situated in Pará State in Brazil, between the coordinates 3°17′23″S and 50°55′8″W, in the Brazilian Legal Amazon (BLA). The area has faced significant deforestation since it started to be monitored by PRODES [20]. Following [47], we used Landsat OLI-8 images acquired on 2 August 2016 (Figure 2a) and 20 July 2017 (Figure 2b).

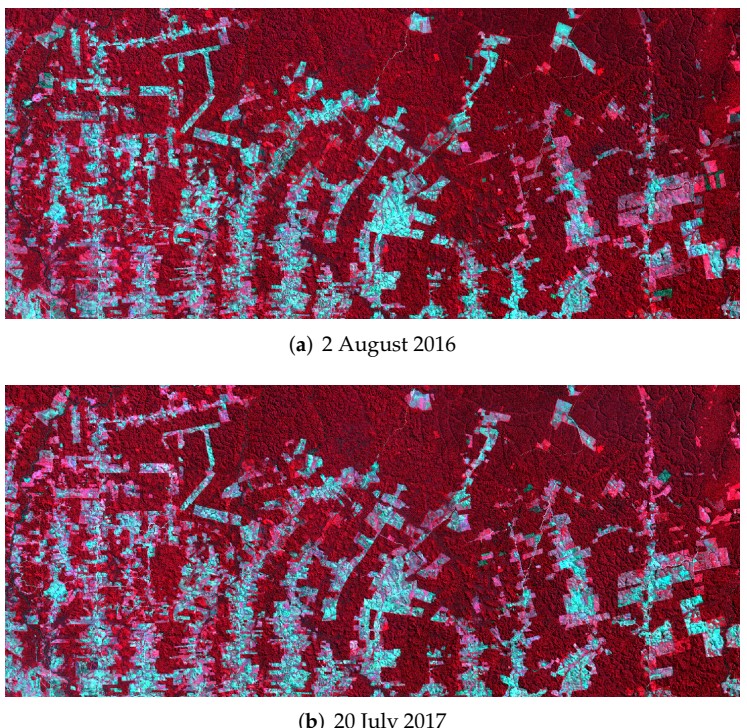

(**a**) 2 August 2016

(**b**) 20 July 2017

**Figure 2.** Landsat 8-OLI false-color composite (bands 5,4,3) of the study area.

The reference deforestation change map, shown in Figure 3, disregards areas deforested before 2016. The data employed in this work was derived from the PRODES database, which is freely available in http://terrabrasilis.dpi.inpe.br/map/deforestation, (accessed on accessed on 9 May 2022). Reference deforestation polygons represent transitions from forest in 2016 to no-forest in 2017, so deforestation class pixels in the reference consist of the intersection between forest in 2016 and no-forest in 2017. As a consequence, remaining pixels are associated with the no-deforestation class, encompassing both areas where the forest cover remains unchanged and native forest were removed prior to 2016. In Figure 3, the areas deforested between 2016 and 2017 are presented in blue while no-deforestation regions are shown in white.

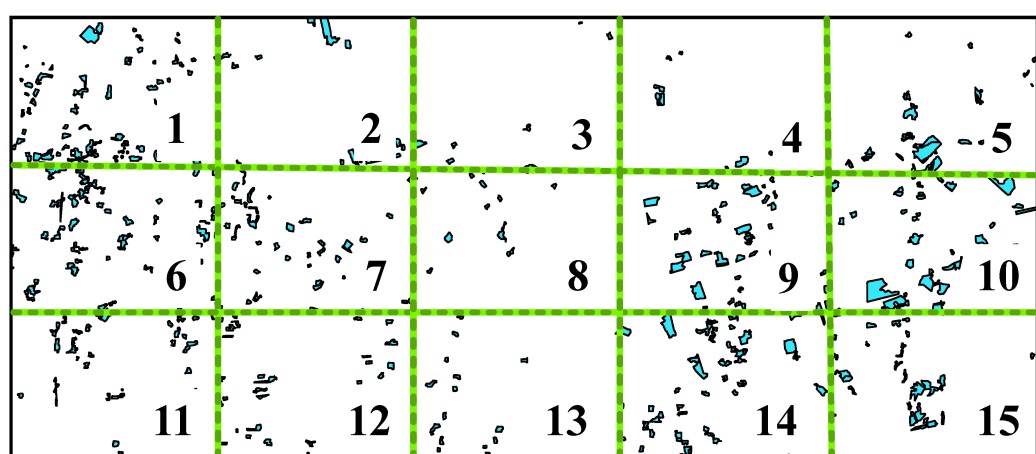

**Figure 3.** Image tiles (numbered) and deforestation polygons (source: [47]).

As shown in Figure 3, the region of interest was divided into 15 tiles, in the form of a 3 × 5 tiles grid. Tiles 1, 7, 9 and 13 were used for training, while tiles 5 and 12 for validation. Tiles 2, 3, 4, 6, 8, 10, 11, 14 and 15 were used exclusively for testing.

In order to evaluate the sensitivity of the evaluated methods to the amount of training data, four different training scenarios were investigated in the experiments: using a single tile for training (tile 13); using two tiles for training (tiles 1 and 13); using three tiles for training (tiles 1, 7 and 13); and using four tiles for training (tiles 1, 7, 9 and 13). Table 1 shows the respective areas covered by the pixels labeled as deforestation within the training tiles used in each training scenario, as well as the proportion of those areas in relation to the total extent of the training tiles.

**Table 1.** Deforestation area and tiles present in the distinct training scenarios.

| Training Scenario | Tiles | Deforestation Area (Pixels) | Deforestation Proportion |
|---|---|---|---|
| single tile | 13 | 2137 | 1.1% |
| two tiles | 1 and 13 | 12,112 | 3.3% |
| three tiles | 1, 7 and 13 | 16,376 | 2.9% |
| four tiles | 1, 7, 9 and 13 | 24,438 | 3.3% |

The dataset comprises a pair of Landsat 8-OLI images, disregarding panchromatic band. Such a 30m spatial resolution image data was subjected to atmospheric correction, and clipped to the target area. The geometric support of the final images was $1100 \times 2600$ pixels. In addition to the seven original spectral bands (Coastal/Aerosol, Blue, Green, Red, NIR, SWIR-1, and SWIR-2), following [47], a Normalized Difference Vegetation Index (NDVI) [62] band was added to each image of the pair.

As mentioned before, the deforestation detection application is characterized by a high class unbalance. Table 2 summarizes the fraction of deforestation area in relation to the total study region area. The training, validation and test set rows in Table 2 show the proportions in relation to the total area covered by the tiles considered in the respective sets.

**Table 2.** Deforestation area in the study region.

| Deforestation | Area (Pixels) | Proportion (%) |
|---|---|---|
| Total | 72,298 | 2.6 |
| Training set | 24,438 | 3.3 |
| Validation set | 8807 | 2.3 |
| Test set | 39,053 | 2.3 |

*5.2. Baseline Methods*

As baselines for the evaluation of DLCD the two *patchwise* semantic segmentation methods, Early Fusion (EF) and Siamese CNN (S-CNN), presented in [47] were selected. Both methods take $15 \times 15$ patches as input and produce a single decision for the entire patch under analysis. The main difference consists in the fact that for EF, seven Landsat 8-OLI bands as well NDVI images for $t_i$ and $t_{i+d}$ time instances are concatenated before been submitted to its input, altogether composing a 3D-tensor with 16 layers. On the hand, S-CNN network has two inputs, where each one receives an 8-layer tensor, one for $t_i$ and the other for $t_{i+d}$.

Table 3 presents a detailed description of the EF and S-CNN architectures employed in this work. In that table, the rows refer to network layers and the columns show the characteristics of such layers. The first column shows layer numbers and types; column 2 presents the output tensor size; while columns 3 to 5 indicate the number and size of kernels, the stride and activation function, when applicable. In the specific case of the EF architecture, there are three convolutional layers interspersed with max-pooling layers. The convolutional layer kernel description contains four numbers. For instance, in the first layer, we have a 128 ($3 \times 3 \times 16$) convolutional layers, that means 128 filters with respectively 3 rows, 3 columns and depth 16. After the last convolutional layer there are two fully connected layers, the first one with 4608 hidden neurons and the last one with two output layers with softmax activation functions.

As it can be noticed in Table 3, S-CNN architecture is quite similar to the EF. The main difference consists in having two independent convolutional-pooling encoders for each time instance. However, as usual in Siamese architectures, both encoders share the same weights' values. Another difference in relation to EF consists of both encoders outputs being concatenated before being submitted as the input of the fully connected layers sequence.

We recall that the EF and S-CNN baseline methods adopt a patch-wise classification approach for dense labeling. Their architectures are, therefore, not fully convolutional, as that of DLCD, having fully connected layers at the end. They take as input one image patch and deliver a single class label for that patch. That label is considered as the prediction for the center pixel position in the output prediction map.

**Table 3.** Detailed description of EF and S-CNN architectures.

| | Early Fusion | | | | |
| --- | --- | --- | --- | --- | --- |
| | **Layer type** | **Output** | **Kernel** | **Stride** | **Activation** |
| | Input Image | $15 \times 15 \times 16$ | - | - | - |
| 1 | Convolutional | $15 \times 15 \times 128$ | $128\ (3 \times 3 \times 16)$ | 1 | ReLU |
| | Max-pooling | $7 \times 7 \times 128$ | $2 \times 2$ | 2 | - |
| 2 | Convolutional | $7 \times 7 \times 256$ | $256\ (3 \times 3 \times 128)$ | 1 | ReLU |
| | Max-pooling | $3 \times 3 \times 256$ | $2 \times 2$ | 2 | - |
| 3 | Convolutional | $3 \times 3 \times 512$ | $512\ (3 \times 3 \times 256)$ | 1 | ReLU |
| 4 | Fully connected | $4608 \times 1$ | - | - | ReLU |
| 5 | Fully connected | $2 \times 1$ | - | - | Softmax |
| | **Total number of parameters** | | | 22,741,378 | |
| | Siamese CNN | | | | |
| | **Layer type** | **Output** | **Kernel** | **Stride** | **Activation** |
| | Input Images (2×) | $15 \times 15 \times 8$ | - | - | - |
| 1 (2×) | Convolutional | $15 \times 15 \times 128$ | $128\ (3 \times 3 \times 8)$ | 1 | ReLU |
| | Max-pooling | $7 \times 7 \times 128$ | $2 \times 2$ | 2 | - |
| 2 (2×) | Convolutional | $7 \times 7 \times 256$ | $256\ (3 \times 3 \times 128)$ | 1 | ReLU |
| | Max-pooling | $3 \times 3 \times 256$ | $2 \times 2$ | 2 | - |
| 3 (2×) | Convolutional | $3 \times 3 \times 512$ | $512\ (3 \times 3 \times 256)$ | 1 | ReLU |
| 4 | Fully connected | $4608 \times 1$ | - | - | ReLU |
| 5 | Fully connected | $2 \times 1$ | - | - | Softmax |
| | **Total number of parameters** | | | 43,970,434 | |

*5.3. Experimental Setup*

The herein employed experimental setup follows the same procedures adopted in [47] and relies on the imagery as well as on the reference data presented in Section 5.1. Following [47], patch sizes for both the EF and S-CNN are $15 \times 15$. During the training cycles, data augmentation was performed only on patches associated with the deforestation class, being each training patch rotated by 90º, and flipped in the horizontal and vertical axis. In addition, in order to balance the training patches, the ones associated with the no-deforestation class were randomly neglected. Thus, in the training scenario with four tiles, 8118 training patches were obtained for each class. The validation set was composed of a total of 40,642 patches, 963 of the deforestation class, and 39,679 of the no-deforestation class. The test set sample numbers also express the natural class distribution, comprising 1,716,000 patches, 40,392 for deforestation class and 1,675,608 for no-deforestation.

The batch size for training the EF and S-CNN methods was set to 32, and early stopping was used to halt training after 10 epochs without improvement. For the last fully connected layer a dropout rate of 0.2 was applied. Training cycles were carried out with the Adam optimizer, using a learning rate of 0.001 and a weight decay of 0.9. The architectures of the EF and S-CNN networks, which were described in Section 5.2, are exactly the same presented in [47].

For the DLCD method image patches of $64 \times 64$ pixels with an overlap of $48 \times 48$ pixels were provided. Training patches were subjected to data augmentation, they were rotated by 90º, 180º, 270º; being both the original and rotated versions flipped vertically. Patches without deforestation pixels were disregarded. The batch size was set to 16, and the maximum number of epochs was set to 100, while early stopping was set to halt the training procedure after 10 epochs without improvement. Training was performed with the Adam optimizer with a learning rate of 0.001.

Balancing the training data for the DLCD method is not so simple. For example, selecting training patches based on any criteria would result in a much smaller training set. Therefore, we dealt with class unbalance by employing the weighted focal loss function [26] during the DLCD model training.

## 6. Results and Discussion

In order to investigate the influence of the weighted focal loss (WFL) function parameters $\alpha$ and $\gamma$ in the classification performance of the DLCD method, we made a grid search. Accordingly, in the experiments, we considered the following values of $\alpha$: 0.9, 0.8, 0.7, 0.6 and 0.5, while, for each different $\alpha$ value, we varied $\gamma$ values from 0 to 5, which amounts to a total of 30 combinations of $\alpha$ and $\gamma$ tuples. The selected $\alpha$ values were intended to cover a range of values that goes from not weighting the classes, i.e., $\alpha = 0.5$, to weighting up to one order of magnitude in favor of the deforestation class, i.e., $\alpha = 0.9$. We recall that the proportion of deforestation pixels in the training tiles varies from 1% to 3%, depending on the training scenario, but as explained in the previous section, only patches with deforestation pixels were selected for training, so we believe that $\alpha = 0.9$ was a good upper bound for that parameter. As for the selected range of $\gamma$ values, we basically adopted the same range used by the authors of [26] (that proposed the WFL). It is important to observe that when $\gamma = 0$, the WFL becomes equivalent to the balanced cross entropy loss. Additionally, with $\gamma = 0$ and $\alpha = 0.5$ the loss degenerates to the ordinary cross entropy loss function, which was the loss employed in the original implementation of DeepLabv3+.

For each tuple of WFL parameters, the mean results of 10 network initialization–training cycles using the four tiles training set scenario are presented in Figure 4. At the top of the figure, the precision average outcomes are presented, below, the bar graph containing average recall values, while average F1-scores are presented in the third chart. The table at the bottom of Figure 4 shows the parameters values employed in each experiment. In addition, the results associated with EF and S-CNN [47], also expressing the mean for 10 initialization–training rounds, were included in that figure after the DLCD results.

Analyzing the precision values in Figure 4, one can notice that the DLCD method trained using WFL produced consistently smaller amounts of false positives than EF and S-CNN. The best DLCD variant, DLCD-29, went beyond S-CNN by more than 25%. However, an analysis of the recall values makes it clear that, on the other hand, EF produced lesser false negatives than the DLCD variants, a behavior that was not observed with S-CNN. This fact can be observed on the recall chart in Figure 4, in which EF outperformed the best among the DLCD evaluated variants, DLCD-3, by approximately 1.5%.

Looking at the results obtained for distinct $\alpha$ values, one can notice that $\alpha = 0.5$ presented by the magenta shaded bars brought fewer false positives (see the precision chart). Nonetheless, it provided more false negatives downgrading the respective recall values. In contrast, $\alpha = 0.9$ (reddish bars) tended to bring superior recall values, suggesting that such $\alpha$ value tends to produce fewer false negatives. On the other hand, that $\alpha$ value produced the poorest precision, providing more false positives.

The observation of the average F1-scores, which stands for the harmonic mean between precision and recall, may bring into the conclusion that, in the performed experiments, F1-scores somehow compensates the conflicting behaviors between precision and recall for the scope of parameter values under analysis. Among the 30 DLCD variants, only three were outperformed by EF and S-CNN in terms of F1-Scores. The best mean F1-score was 73.42% obtained for DLCD-14 which used $\alpha = 0.7$ and $\gamma = 1$, outperforming EF in more than 10%. That outcome suggests a clear superiority of the herein presented method in terms of F1-scores.

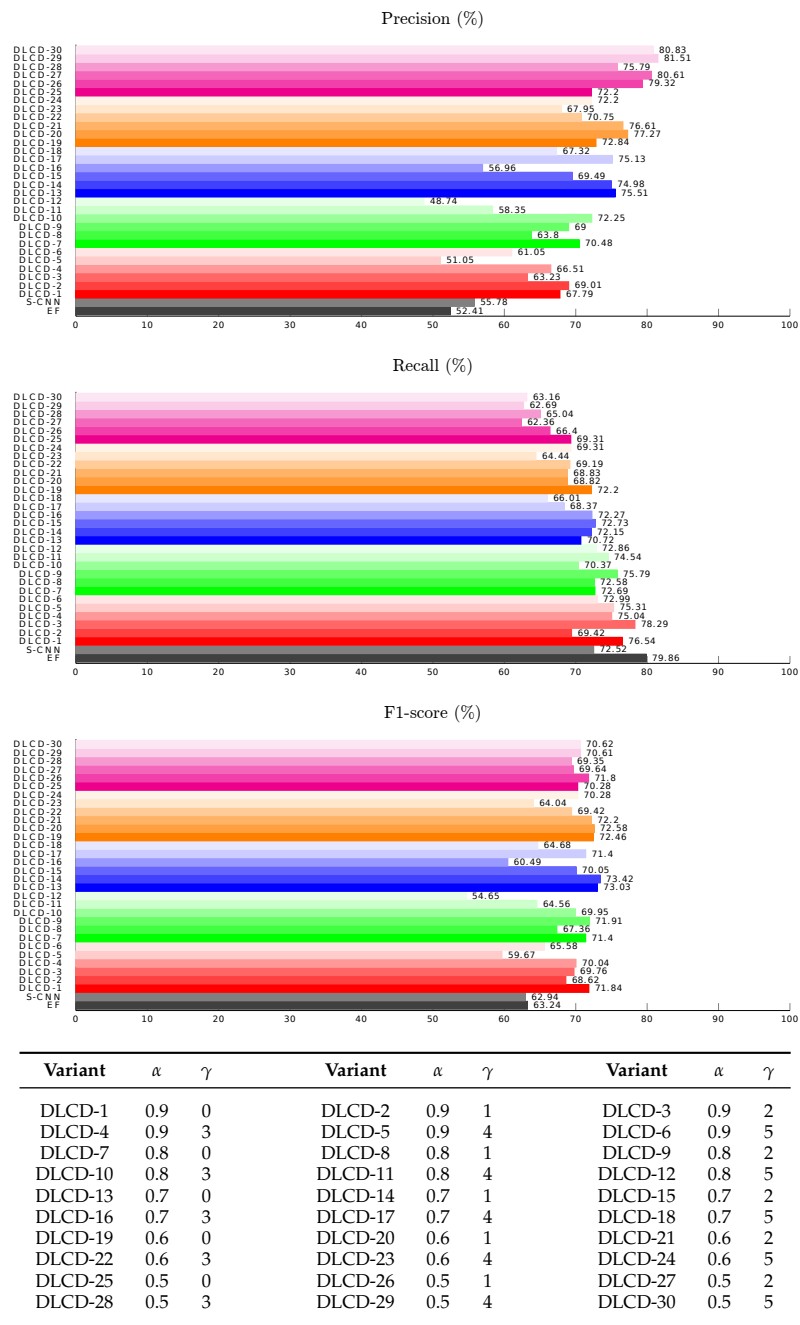

| Variant | $\alpha$ | $\gamma$ | Variant | $\alpha$ | $\gamma$ | Variant | $\alpha$ | $\gamma$ |
|---------|----------|----------|---------|----------|----------|---------|----------|----------|
| DLCD-1 | 0.9 | 0 | DLCD-2 | 0.9 | 1 | DLCD-3 | 0.9 | 2 |
| DLCD-4 | 0.9 | 3 | DLCD-5 | 0.9 | 4 | DLCD-6 | 0.9 | 5 |
| DLCD-7 | 0.8 | 0 | DLCD-8 | 0.8 | 1 | DLCD-9 | 0.8 | 2 |
| DLCD-10 | 0.8 | 3 | DLCD-11 | 0.8 | 4 | DLCD-12 | 0.8 | 5 |
| DLCD-13 | 0.7 | 0 | DLCD-14 | 0.7 | 1 | DLCD-15 | 0.7 | 2 |
| DLCD-16 | 0.7 | 3 | DLCD-17 | 0.7 | 4 | DLCD-18 | 0.7 | 5 |
| DLCD-19 | 0.6 | 0 | DLCD-20 | 0.6 | 1 | DLCD-21 | 0.6 | 2 |
| DLCD-22 | 0.6 | 3 | DLCD-23 | 0.6 | 4 | DLCD-24 | 0.6 | 5 |
| DLCD-25 | 0.5 | 0 | DLCD-26 | 0.5 | 1 | DLCD-27 | 0.5 | 2 |
| DLCD-28 | 0.5 | 3 | DLCD-29 | 0.5 | 4 | DLCD-30 | 0.5 | 5 |

**Figure 4.** Precision Recall and F1-Scores obtained for distinct Weighted Focal Loss function parameters while using the four tiles training set scenario.

Table 4 aggregates the results shown in Figure 4. In an attempt of isolating the impact of the $\gamma$ parameter, Table 4 presents the average values obtained for distinct $\gamma$ values disregarding $\alpha$. By analyzing the results presented in Table 4, one can notice that the

highest F1-Score, Recall and Precision were obtained when $\gamma$ is equal to zero. This result suggests that it is possible to obtain stable results when WFL is reduced to weighted cross-entropy (WCE). However, the choice of $\gamma$ value should take into account that the best F1-score in Figure 4 was 73.42%, a result obtained with DLCD-14, which employed $\alpha = 0.7$ and $\gamma = 1$. Anyway, in general terms, increasing $\gamma$ to values beyond two demonstrates a noticeable impact on F1-Score, Precision and Recall. However, it should be noticed that the decay on Precision outweighs that on Recall. That result suggests that false positives (false deforestation alarm) are even more impacted by the increment of $\gamma$ than false negatives.

**Table 4.** Mean of F1-Scores, Recall, and Precision disregarding $\alpha$ and grouped by $\gamma$.

| $\gamma$ Value | F1-Score | Recall | Precision |
|:---:|:---:|:---:|:---:|
| 0 | 71.80 | 72.29 | 71.76 |
| 1 | 70.76 | 69.87 | 72.87 |
| 2 | 70.71 | 71.60 | 71.79 |
| 3 | 67.85 | 70.38 | 68.45 |
| 4 | 66.05 | 69.07 | 66.80 |
| 5 | 65.16 | 68.87 | 66.03 |

In another set of experiments, we sought to evaluate the impact of training set size on DLCD as well as on EF and S-CNN. For that purpose, Figure 5 presents F1-scores provided while using distinct numbers of tiles to generate the training data. Hence, the more tiles employed, the larger is the training set size. For all number of tiles selected DLCD-14 performed better than both EF and S-CNN. Comparing S-CNN to EF, S-CNN presented better results for 1, 2 and 3 tiles, while EF preformed better for a training set generated from 4 image tiles. However, the most important fact to be observed is that the smaller the training set was, the greater the advantage of DLCD in relation to EF and S-CNN. That may lead to the conclusion that the use of the DLCD method is even more suitable in cases where the training set is restricted. It should be stressed that this is a very important result, since, in remote sensing change detection, the limitation of the amount of training data is a quite common condition. Thus, considering such an advantage, the use of DLCD should be preferable to EF and S-CNN.

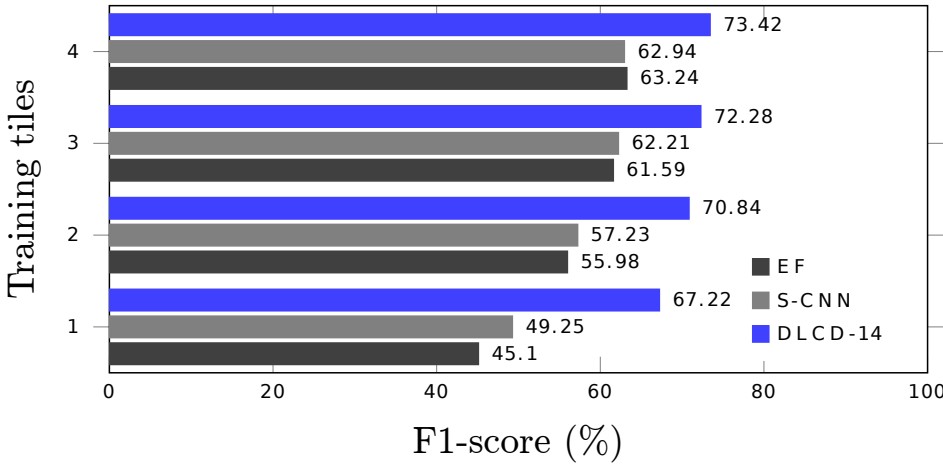

**Figure 5.** F1-Scores for distinct volumes of training data.

Figure 6 presents the change maps produced by EF, S-CNN, and DLCD-14 for tiles 6 and 14, between the years of 2016 and 2017. The left column shows the results for tile 6, and the right column shows the outcomes for tile 14. The top row presents results produced by EF, the middle row shows the ones produced by S-CNN, and the bottom row show the results obtained with DLCD-14. Comparing those maps, it is possible to notice that the proposed method produced a slightly larger amount of false no-deforestation

errors (plotted in blue) than EF and S-CNN. On the other hand, DLCD produced a notably lower number of false deforestation errors (indicated in red). We observe that such an outcome is particularly important for operational reasons, considering the effort and costs involved in the reconnaissance of the actual deforestation by the local authorities, involved in penalizing the perpetrators or in mitigating the effects of illegal deforestation.

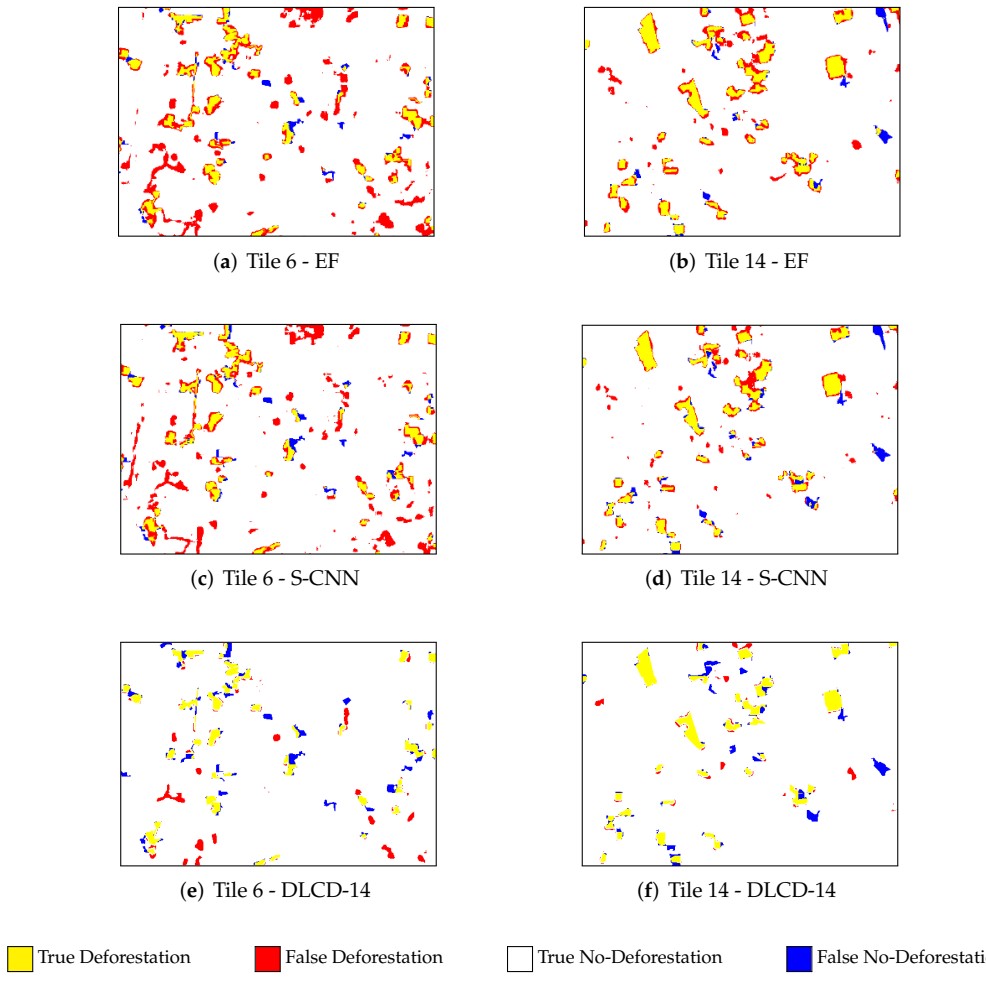

(**a**) Tile 6 - EF

(**b**) Tile 14 - EF

(**c**) Tile 6 - S-CNN

(**d**) Tile 14 - S-CNN

(**e**) Tile 6 - DLCD-14

(**f**) Tile 14 - DLCD-14

▨ True Deforestation     ▮ False Deforestation     ▯ True No-Deforestation     ▮ False No-Deforestation

**Figure 6.** Change maps predicted by EF, S-CNN, and DLCD-14 on test tiles 6 and 14 for 2016–2017.

To close this section we observe that, although the results obtained with the proposed DLCD method for the majority of the $\alpha$ and $\gamma$ value combinations produced better accuracies in terms of F1-scores than the baseline methods, in a few combinations, i.e., 0.7, 3; 0.8, 4; and 0.9, 4, the accuracy values were slightly inferior. Additionally, the differences in accuracy values between the best and the worst hyperparameter combinations vary substantially (up to about 19% in F1-score). Therefore, the selection of proper $\alpha$ and $\gamma$ values is key to obtain superior results. Additionally, although we designed the experiments in a way to avoid a high level of spatial correlation between the training and test samples (by extracting them from different image tiles), we cannot be sure that the optimum $\alpha$ and $\gamma$ values found for the target site are also the best for other sites, with dissimilar forest covers. Such generalization issues open grounds for further investigation.

## 7. Conclusions

In this work, we proposed and evaluated a deep-learning-based deforestation detection method. The method was built on top of the DeepLabv3+ architecture. The original DeepLabv3+ model was adapted in several ways and we adopted the weighted focal loss

function during training, aiming at improving the accuracy of the detected deforestation polygons silhouettes, and at dealing with the extreme class unbalance, characteristic of the deforestation detection application. The proposed method, denoted DeepLab Change Detection (DLCD), was compared with two baseline methods: Early Fusion (EF), Siamese Convolutional Neural Network (S-CNN), which belong to the current state-of-the-art methods for the deforestation detection application.

The experiments addressed the analysis of the impact of the training set size on the method's performance. We carried out an extensive grid search on the weighted focal loss function hyperparameters in the DLCD training procedure. When compared to the baseline methods, despite presenting smaller Recall values, variants of the proposed method significantly outperformed the EF and S-CNN methods in terms of Precision and F1-score. In addition, for smaller amounts of training data, more significant performance gains were presented by the DLCD method. That was an important result, since limited amounts of training data is a reality in many change detection problems. Another significant advantage of the proposed method is that, being a dense labeling model, it provides considerably lower inference times than patchwise classification methods like S-CNN and EF. Additionally, DLCD produced a notable lower number of false positive errors, which is crucial for such an application, considering the effort and costs involved to reinforce environmental laws in a vast region like the Brazilian Legal Amazon.

The adoption of the weighted focal loss has proven effective in dealing with the characteristic class unbalance of the target application. Nonetheless, the use of such loss function introduced some additional parameters when compared with S-CNN and EF. The herein presented evaluation indicates that $\gamma \in \{0,1\}$ and $\alpha \in [0.6, 0.7]$ may be good options in future works in order to fasten the fine-tuning of such parameters.

In regard to directions for future work, we plan to investigate if the performance of the proposed semantic segmentation approach is transferable to other domains, e.g., training the proposed model with samples from a particular site, and testing on images covering different forest sites. In fact, investigating deep learning based domain adaptation is planned for the continuation of this research. Another possibility is adapting recent image classification approaches not yet applied to this problem ([36,38,39] for instance) for the deforestation detection problem.

**Author Contributions:** Conceptualization, G.A.O.P.d.C. and G.L.A.M.; methodology, G.A.O.P.d.C. and R.B.d.A.; software, R.B.d.A.; formal analysis, G.A.O.P.d.C. and G.L.A.M.; investigation, R.B.d.A.; data curation, R.B.d.A.; writing—original draft preparation, G.L.A.M.; writing—review and editing, G.A.O.P.d.C. and G.L.A.M.; supervision, G.A.O.P.d.C. and G.L.A.M.; project administration, G.A.O.P.d.C. and G.L.A.M.; funding acquisition, G.A.O.P.d.C. and G.L.A.M. All authors have read and agreed to the published version of the manuscript.

**Funding:** This research was funded by CAPES (Brazillian Coordination for the Improvement of Higher Education Personnel) and FAPERJ (Rio de Janeiro State Research Funding Agency).

**Data Availability Statement:** All data employed in this research can be publicly downloaded at http://terrabrasilis.dpi.inpe.br/downloads/ (accessed on 9 May 2022).

**Acknowledgments:** Authors want to express their gratitude to the team of the LVC lab. at Pontifical Catholic University of Rio de Janeiro for the technical support and cooperation in this research.

**Conflicts of Interest:** The authors declare no conflict of interest.

## Abbreviations

The following abbreviations are used in this manuscript:

| | |
|---|---|
| ASPP | Atrous-convolutional Spatial Pyramid Pooling |
| CD | Change Detection |
| CNN | Convolutional Neural Networks |
| Concat | Feature Mapas Concatenation |
| Conv | Regular 2D Convolution |
| CRF | Conditional Random Fields |
| DL | Deep Learning |
| EF | Early Fusion |
| FCN | Fully Convolutional Networks |
| INPE | Brazilian National Institute for Space Research |
| MLP | Multi Layer Perceptron |
| NDVI | Normalized Difference Vegetation Index |
| ReLU | Rectified Linear Unit activation function |
| RS | Remote Sensing |
| SConv | Depthwise Separable Convolution |
| SS | Semantic Segmetation |
| SVM | Suport Vector Machine |
| WFL | Weighted Focal Loss |

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
