# Peer review of "Deforestation Detection in the Amazon Using DeepLabv3+ Semantic Segmentation Model Variants"

_remotesensing, doi:10.3390/rs14194694_

Round 1

Reviewer 2 Report

1. In abstract, please provide values for the last sentence.

2. Conclusion design is not correct. Authors should provide a shorter version and most of the Conclusion content (including the Figure) to Results. 

3. In results (maybe an additional discussion) section. Authors should discuss model limitations, possible drawbacks etc.  

4. Figure 2 image contrast and quality is not so good. Authors can try 7,5,3 (SWIR 2, NIR, Green) band combination for a better view of a forested area.

Reviewer 3 Report

This paper proposed to detect the Deforestation in the Amazon using the DeepLabV3+ Semantic Segmentation networks. The idea is interesting and potential, and the experimental results are competitive. I only have some minor issues that need to be fixed.

1. A deep literature reviews should be given, particularly advanced and SOTA deep learning or AI models in remote sensing image classification. Therefore, the reviewer suggests discussing some related works by analyzing the following papers in the revised manuscript, e.g., Graph convolutional networks for hyperspectral image classification, More diverse means better: Multimodal deep learning meets remote-sensing imagery classification, Hybrid Dense Network With Attention Mechanism for Hyperspectral Image Classification

2. Please clarify the contributions to this field, for example, which are the existing ones and which are your own ones?

3. What are the differences in techniques between the proposed method and existing methods?

4. Some future directions should be pointed out in the conclusion.

Round 2

Reviewer 3 Report

The references in the revised manuscript are missing, there are some typos in the mansucript.
